# Presence of Concurrent *TP53* Mutations Is Necessary to Predict Poor Outcomes within the *SMAD4* Mutated Subgroup of Metastatic Colorectal Cancer

**DOI:** 10.3390/cancers14153644

**Published:** 2022-07-27

**Authors:** Chongkai Wang, Jaideep Sandhu, Amber Tsao, Marwan Fakih

**Affiliations:** Department of Medical Oncology and Therapeutics Research, City of Hope Comprehensive Cancer Center, Duarte, CA 91010, USA; chowang@coh.org (C.W.); jsandhu@coh.org (J.S.); ambertsao@gmail.com (A.T.)

**Keywords:** colorectal cancer, *SMAD4*, *TP53*, prognosis

## Abstract

**Simple Summary:**

Prior studies have resulted in conflicting conclusions on the value of *SMAD4* mutations as a prognostic biomarker in metastatic colorectal cancer. In a cohort study of 433 patients with metastatic colorectal cancer, we showed that the presence of a coexisting mutation in *TP53* is necessary to culminate in a negative overall survival impact in patients with *SMAD4* mutations (multivariate HR = 2.5, 95% CI 1.44–4.36, *p* = 0.001). Our findings indicate that patients with concurrent *SMAD4* and *TP53* mutations represent a distinct poor-prognosis subgroup that may benefit from further translational studies.

**Abstract:**

Prior studies have resulted in conflicting conclusions on the value of *SMAD4* mutations as a prognostic biomarker in metastatic colorectal cancer. In this study, the impact of coexisting mutations with *SMAD4* on overall survival was evaluated retrospectively in 433 patients with metastatic colorectal cancer. *SMAD4* mutation was found in 16.2% (70/433) of tumors. A systemic univariate and multivariate survival analysis model including age, gender, sidedness of primary tumor, *RAS*, *BRAF^V600E^*, *APC*, *TP53* and *SMAD4* status showed that *SMAD4* mutations were not associated with worse prognosis (multivariate HR = 1.25, 95% CI 0.90–1.73, *p* = 0.18). However, coexisting mutations in *SMAD4* and *TP53* were significantly associated with worse overall survival (multivariate HR = 2.5, 95% CI 1.44–4.36, *p* = 0.001). The median overall survival of patients with coexisting *SMAD4* and *TP53* mutation was 24.2 months, compared to 42.2 months for the rest of the population (*p* = 0.002). Concurrent *SMAD4* and *TP53* defines a new subgroup of patients of metastatic colorectal cancer with poor clinical outcomes.

## 1. Introduction

With 151,300 new cases and 52,580 deaths estimated in 2022, colorectal cancer remains the second most common cause of cancer-related death in the United States. While localized disease has a 5-year overall survival rate of 90.9%, only 15% of patients with distant metastatic disease can survive for 5 years [1]. Colorectal cancer is a highly heterogeneous disease with diverse genetic and pathologic features that contribute to differences in treatment response and, ultimately, overall survival [2,3]. Identification of tumor biomarkers capable of matching patients to appropriate targeted therapies has demonstrated great success, as reflected by the impact of sidedness and *RAS/BRAF* status in selecting patients for anti-EGFR treatment, and by the value of microsatellite instability in identifying appropriate patients for PD-1 inhibition [4,5]. A comprehensive molecular tumor mapping is needed to better prognosticate and guide the clinical management of patients with metastatic colorectal cancer.

*SMAD4*, a tumor suppressor, is a key mediator of the TGF-β signaling pathway [6]. Sporadic *SMAD4* mutation is found in approximately 10–20% of colorectal cancers [7,8]. Prior studies have shown that loss of *SMAD4* is associated with worse recurrence-free and overall survival in patients with stage III colorectal cancer [9]. In addition, studies have shown that *SMAD4* loss is correlated with resistance to 5-fluorouracil-based chemotherapy [10,11]. In patients with metastatic colorectal cancer, some studies have correlated *SMAD4* mutation with worse overall survival, while others could not confirm such a finding [8,12]. Recently, studies have demonstrated that analyzing coexisting mutations in multiple genes is better at predicting prognosis than in a single gene alone. In patients with resection of colorectal liver metastases, concurrent *RAS* and *TP53* mutations were associated with worse overall survival (OS) than either mutation alone [13]. This finding was confirmed in other studies, where co-mutation of *RAS/BRAF* and *TP53* was correlated with significantly worse OS in metastatic colorectal cancer to the liver following complete metastasectomy [14,15]. We hypothesized that the co-occurrence of frequently mutated oncogenes with *SMAD4* would be more valuable as a prognostic biomarker than *SMAD4* alone in patients with metastatic colorectal cancer.

## 2. Materials and Methods

### 2.1. Patient Population

Patients with metastatic colorectal cancer treated at the City of Hope Comprehensive Cancer Center (Duarte, CA, USA) between 2013 and 2020 with available tumor NGS by a CLIA-certified assay were eligible for this study. A total of 433 patients were identified for this study (Figure 1). Patients’ characteristics, including age, gender, sidedness, and survival status, were obtained from a chart review of each patient’s electronic medical record. All patients received standard of care. This study was approved by the Institutional Review Board IRB 14361.

### 2.2. Genomic Analysis

Comprehensive genomic profiling of tumor samples was conducted through a CLIA-certified NGS platform via FoundationOne (*n* = 339, Foundation Medicine Inc., Cambridge, MA, USA), GEM ExTra (*n* = 11, Ashion Analytics, Phoenix, AZ, USA), and HOPESEQ (in-house platform, *n* = 15). In total, 68 patients were included based on the genomic profile from liquid biopsy (Guardant 360, Guardant Health, Redwood City, CA, USA). The genomic profile of each tumor was extracted from the NGS report of these assays.

### 2.3. Statistical Analysis

Patients’ characteristics and genomic alterations were analyzed by the Wilcoxon rank test (age) and Fisher’s exact test (categorical variables). OS was defined as the time between the date of first evidence of metastatic disease and the date of death. Differences in OS were compared using Kaplan–Meier curves, with *p*-values calculated via log-rank test. Univariate and multivariate Cox regression models were used to evaluate the hazard ratios of overall survival based on mutational status and other factors. IBM SPSS Version 28.0.1.1 (15) was used for univariate and multivariate analysis. GraphPad Prism 9.3.1 was used for the baseline characteristic table.

## 3. Results

### 3.1. Study Population

A total of 433 stage IV colorectal adenocarcinomas were included in this study. Median age was 55 years, 42% were female (*n* = 182), and 58% were male (*n* = 251). Liver, lung, lymph node, and peritoneal metastasis were found in 64.8% (*n* = 281), 24.1% (*n* = 104), 22.4% (*n* = 97), and 17.8% (*n* = 77) of the patients. Left-sided tumors were found in 72% of the patients (*n* = 309). *SMAD4* mutation (*SMAD4*-MT) was found in 16.2% (70/433) of tumors. Four genes had a somatic mutation rate higher than 10% in the *SMAD4* mutated population: *APC* (59%, *n* = 41), *BRAF^V600E^* (11%, *n* = 8), *RAS* (54%, *n* = 38), and *TP53* (80%, *n* = 56). Wild-type *APC* was found in 41% of *SAMD4*-MT tumors and 26% of *SMAD4* wild-type (*SMAD4*-WT) tumors (*p* = 0.01). No significant difference in age, gender, sidedness, *TP53, RAS*, and *BRAF^V600E^* mutation was observed between *SMAD4*-MT and *SMAD4*-WT populations (Table 1).

### 3.2. SAMD4 Mutation Alone Does Not Correlate with Poor Prognosis

The overall median follow-up was 61 months. A systemic univariate survival analysis model including age, gender, sidedness of primary tumor, *RAS*, *BRAF^V600E^*, *APC*, *TP53*, and *SMAD4* status showed that right-sided primary tumor, *BRAF^V600E^* mutation, and wild-type *APC* were associated with worse prognosis (*p* < 0.05) (Appendix A). The statistical significance was lost in our multivariate analysis including the same variables (Table 2). The median overall survival of *SMAD4*-MT patients was 28.5 months, compared to 41.2 months in patients with *SMAD4*-WT (*p* = 0.095) (Figure 2A).

### 3.3. Co-Mutation of SMAD4 with TP53 Is Associated with Worse Overall Survival

Because our analysis did not confirm prior findings that *SMAD4* mutation alone was associated with worse prognosis, we evaluated whether co-mutations may contribute to the prognosis of patients with *SMAD4* mutation. A univariate survival analysis model including the most frequent concurrent mutations, such as *RAS*, *BRAF^V600E^*, *APC*, and *TP53*, in the *SMAD4*-MT group showed that coexistence of *TP53* mutation was significantly associated with worse prognosis (HR = 2.74, 95% CI 1.15–6.52, *p* = 0.02) (Appendix A). Using these genes in a multivariate survival analysis confirmed that concurrent mutation of *TP53* was predictive of worse prognosis in patients with *SMAD4* mutation (HR = 3.2, 95% CI 1.31–7.8, *p* = 0.01) (Table 3). In contrast, *TP53* mutation has no significant impact on OS in patients with wild-type *SMAD4* (Appendix A). To rule out the possibility that our data analysis was driven by the comparison of the *SMAD4*-MT/*TP53*-MT group to a select of *SMAD*-MT/*TP53*-WT cohort with a good prognosis, we calculated multivariable HRs for double mutations in comparison to the rest of the overall population. Similarly, coexistence of mutations in *TP53* with *SMAD4* was significantly associated with worse OS than the rest of the overall population (HR = 2.5, 95% CI 1.44–4.36, *p* = 0.001) (Table 4). Co-mutation of *RAS* or *BRAF^V600E^* with *SMAD4* did not show any significant impact on OS in our analysis (HR = 0.95, *p* = 0.853; HR = 1.5, *p* = 0.353, respectively). Interestingly, patients with co-mutation of *APC* and *SMAD4* had a better OS than the rest of the population (HR = 0.51, 95% CI 0.29–0.91, *p* = 0.022) (Table 4). The median OS of *SMAD4/TP53* mutated metastatic colorectal cancer patients was 24.2 months, compared to 42.2 months for the rest of the population (*p* = 0.0017) (Figure 2B).

Given that prior studies have shown that the addition of *TP53* mutation makes prognosis worse in patients with *RAS* mutations, we also investigated the impact of *RAS*/*TP53* mutation on prognosis in our cohort. Patients with co-mutations of *RAS* and *TP53* had a worse OS than the rest of the population (median OS, 33.8 vs. 45.2 months, *p* = 0.035, Appendix A). However, the significance of *RAS*-MT and *TP53*-MT on OS was lost in our multivariate analysis (Table 4). Taken together, our results showed that coexisting mutation in *SMAD4* and *TP53* identified a unique subgroup of patients with poor prognosis.

## 4. Discussion

In this era of personalized medicine for patients with colorectal cancer, uncovering prognostic and predictive biomarkers is paramount for treatment selection and outcome prediction. While some prior studies have correlated *SMAD4* mutation with a worse prognosis, others did not confirm such findings. However, these studies did not evaluate the impact of coexisting mutations in the *SMAD4*-MT population. The strength of our study is the comprehensive evaluation of the four most highly mutated genes, to stratify prognosis in patients with metastatic colorectal cancer harboring *SMAD4* mutation. We showed, in this study, that *SMAD4* mutation alone is not a sufficient biomarker for predicting prognosis in patients with colorectal cancer. Among genes including *APC*, *RAS*, *BRAF^V600E^*, and *TP53*, our analysis showed that the coexistence of *TP53* and *SMAD4* mutation was associated with a significantly worse prognosis in patients with metastatic colorectal cancer.

*SMAD4* is a key component of the SMAD family that modulates signals from the transforming growth factor β (TGF-β) pathways [16,17]. Studies have shown that loss of *SMAD4* promotes TGF-β-mediated tumorigenesis by abrogating the tumor-suppressive functions of TGF-β, such as cell cycle arrest, while maintaining the tumor-promoting functions of TGF-β, such as epithelial–mesenchymal transition [18]. Several studies have correlated the absence of *SMAD4* to worse prognosis in patients with early-stage colorectal cancer [19,20,21]. In addition, *SMAD4* loss in colorectal cancer has been associated with higher tumor and nodal stage, shorter relapse-free survival in 5-FU-based chemotherapy, and less immune infiltration in the tumor microenvironment [10,11]. While our results showed that patients with *SMAD4* mutations had a numerically shorter OS than patients with wild-type *SMAD4*, such a difference did not reach statistical significance in our analysis.

*TP53* is a tumor suppressor that prevents cells from progressing though the cell cycle in response to DNA damage [22]. Studies have shown inconsistent results on the predictive and prognostic value of *TP53* mutation in colorectal cancer. Some studies have suggested that *TP53* is not a prognostic but rather a predictive biomarker for colorectal cancer. A study of 18,766 patients with colorectal cancer showed that *TP53* mutation had no effect on outcomes in patients treated with chemotherapy [23]. In patients with resectable colorectal liver metastases, *TP53* mutations had no effect on overall survival when treated with surgery alone but caused significantly inferior survival in patients receiving neoadjuvant chemotherapy [24]. In studies with stage III colorectal cancer, patients with *TP53* mutation derived less benefit from chemotherapy [25,26]. Recently, co-mutation of *TP53* with other oncogenes such as *RAS* or *BRAF* has been associated with significantly worse prognosis than mutations in *RAS* or *TP53* alone in patients undergoing curative-intent metastasectomy of metastatic colorectal cancer [15]. In addition, the coexistence of *RAS* and *TP53* mutations caused a moderately worse prognosis than either gene alone in patients with metastatic colorectal cancer (FOCUS trial) [27]. These results suggest that there is a potential synergistic deleterious effect between *TP53* and other oncogenic mutations in cancer progression.

A recent study by Pan et al. classified *TP53* mutations into gain-of-function (GOF) and non-gain-of-function (non-GOF), and showed that *TP53*-GOF mutation was associated with worse OS in patients with left-sided but not in right-sided metastatic colorectal cancer [28]. To confirm this finding and also explore whether *TP53*-GOF mutation exerts a greater effect on the prognosis of patients with *SMAD4* mutation, we stratified *TP53* mutation into GOF and non-GOF based on the same criteria. Our analysis did not find any significant difference in OS between *TP53*-GOF and *TP53* non-GOF in patients with left-sided and right-sided metastatic colorectal cancer (Appendix A). Our study showed that patients with co-mutations in *SMAD4* and *TP53*-GOF had better OS than patients with co-mutations in *SMAD4* and *TP53*-non-GOF (median OS 51.1 vs. 17.9 months, *p* = 0.02) (Appendix A). Given the lack of a validation cohort in Pan’s study and the limited sample size of patients with concurrent *SMAD4* and *TP53*-GOF mutation in our cohort, the impact of *TP53*-GOF mutation on the prognosis of metastatic colorectal cancer remains to be determined.

Our data do not provide a mechanistic explanation on how mutant *TP53* and *SMAD4* interact, although crosstalk between the TGF-β and *TP53* pathways has been reported by multiple studies [29,30]. A preclinical study showed that TGF-β-induced tumor cell migration and invasion are empowered by mutant *TP53*, but not wild-type *TP53* [31]. Considering that SMAD4 is the key component of the TGF-β pathway, it may be that mutant *TP53* exerts a deleterious effect on the biological behavior of tumor cells harboring *SMAD4* mutation; thus, the hazard for survival increases from single mutation to double mutation. Our data showed that *TP53* mutations only affect the prognosis of tumors with *SMAD4* mutation and not those with wild-type *SMAD4,* further supporting our hypothesis.

Another interesting finding of our study was that coexisting mutations in *SMAD4* and *APC* had a lower hazard rate of death compared to the rest of the population. This could be driven by the poor prognostic impact of the *APC*-WT tumors, as previously reported by our group [32]. *BRAF*^V600E^ was associated with worse OS in the univariate analysis of the whole population but lost its significance in the multivariate analysis, which suggests that the prognostic value of *BRAF*^V600E^ is affected by co-variables, including *SMA4* and *TP53*. Indeed, these findings add more weight to our findings. It is also possible that that *BRAF-^V600E^* may have been partially confounded by the small subpopulation with MSI (up to 30% of *BRAF-^V600E^* patients may be MSI).

## 5. Limitations

This study is limited by its retrospective design. However, the large sample size provides strong support to our findings. Mutational information was extracted from NGS reports performed on either tumor tissue or liquid biopsy depending on availability. We do not believe that this is likely to have impacted the results as many studies have shown a high degree of concordance for mutations between tumors and circulating tumor DNA [33,34]. The analysis of tumor mutations with prevalence in less than 10% of patients with *SMAD4* mutation was not included in our analysis as our sample size was not large enough to assess the impact of low-frequency alterations. The genes that we included, *RAS*, *BRAF*, *APC*, *SMAD4*, and *TP53*, are driver mutations in major oncogenic pathways and have been implicated as prognostic or predictive biomarkers in prior metastatic colorectal cancer studies. Given that this was a retrospective study and given the heterogeneity of treatment and tumor staging across different lines of treatment, we did not analyze the impact of *SMAD4* and *TP53* co-mutation on progression-free survival or response rate.

## 6. Conclusions

In conclusion, coexisting mutations in *SMAD4* and *TP53* are associated with worse overall survival than concurrent *TP53* wild-type *SMAD4* mutation patients with metastatic colorectal cancer. Our findings suggest that patients with concurrent *SMAD4* and *TP53* mutation represent a distinct prognostic subgroup that may benefit from further translational study, and this should be considered as a stratification biomarker in future prospective trials in metastatic colorectal cancer.

## Figures and Tables

**Figure 1 cancers-14-03644-f001:**
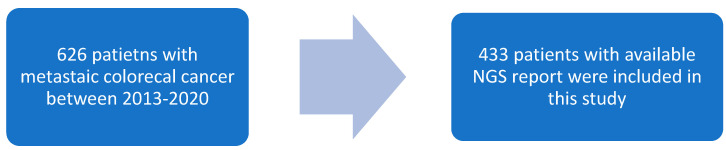
626 patients with colorecal cancer was identified between 2013–2020. 433 of them had NGS report availabe therefore were eligible for this study.

**Figure 2 cancers-14-03644-f002:**
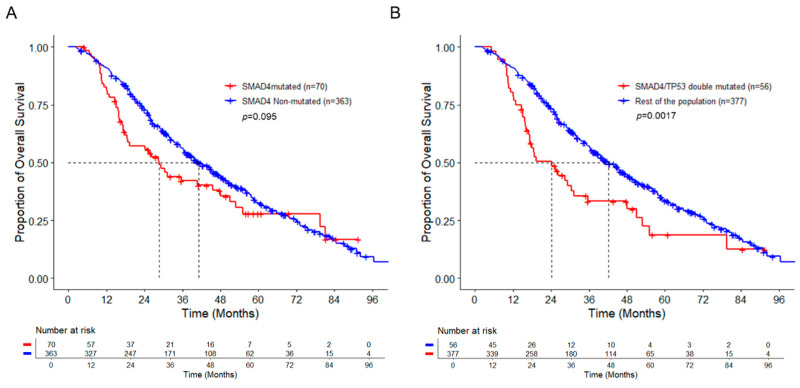
Kaplan–Meier curves for overall survival of metastatic colorectal cancer patients by *SMAD4* and *TP53* status. (**A**), Red line, Kaplan–Meier curve of overall survival for patients with *SMAD4* mutation; Blue line, Kaplan–Meier curve of overall survival for patients without *SMAD4* mutation. (**B**), Red line, Kaplan–Meier curve of overall survival for patients with *SMAD4* and *TP53* double mutation; Blue line, Kaplan–Meier curve of overall survival for patients without *SMAD4/TP53* double mutation.

**Table 1 cancers-14-03644-t001:** Characteristics of patients with metastatic colorectal cancer.

Characteristics		Total (%)		*SMAD4*-WT		*SMAD4*-MT		*p*-Value
		(*n* = 433)		(*n* = 363)		(*n* = 70)		
Age at Diagnosis (Median, Range)		55 (16–90)		55 (20–90)		56 (16–77)		0.78
Gender								
	Male	251	58%	212	58%	39	56%	0.69
	Female	182	42%	151	42%	31	44%	
Sidedness *								
	Left	309	72%	264	73%	45	64%	0.19
	Right	121	28%	97	27%	24	34%	
*RAS*								
	Mutated	207	48%	169	47%	38	54%	0.24
	Non-mutated	226	52%	194	53%	32	46%	
*BRAF* ^V600E^								
	Mutated	32	7%	24	7%	8	11%	0.21
	Non-mutated	401	93%	339	93%	62	89%	
*APC*								
	Mutated	311	72%	270	74%	41	59%	0.01
	Non-mutated	122	28%	93	26%	29	41%	
*TP53*								
	Mutated	346	80%	290	80%	56	80%	1.00
	Non-mutated	87	20%	73	20%	14	20%	

Abbreviations: WT, wild-type; MT, mutated. * Denotes three missing values.

**Table 2 cancers-14-03644-t002:** Multivariate survival model analysis for OS in 433 patients.

Clinicopathogenic Variables	COH Cohort (*n* = 433)			
		95% CI		
	HR	Lower	Upper	*p*-Value
Age at diagnosis (years)				
≥65 vs. <65	0.98	0.74	1.30	0.88
Gender				
Male vs. Female	1.19	0.93	1.52	0.17
Sidedness				
Right vs. Left	1.33	0.99	1.79	0.06
*RAS*				
Mutated vs. Non-mutated	1.23	0.95	1.60	0.11
*BRAF* ^V600E^				
Mutated vs. Non-mutated	1.43	0.86	2.39	0.17
*APC*				
Mutated vs. Non-mutated	0.76	0.58	1.01	0.06
*TP53*				
Mutated vs. Non-mutated	1.36	0.99	1.85	0.06
*SMAD4*				
Mutated vs. Non-mutated	1.25	0.90	1.73	0.18

**Table 3 cancers-14-03644-t003:** Multivariate survival model analysis for OS in patients with *SMAD4* mutation.

Clinicopathogenic Variables	*SMAD4*-MT Cohort (*n* = 70)			
		95% CI		
	HR	Lower	Upper	*p*-Value
*RAS*				
Mutated vs. Non-mutated	1.06	0.58	1.96	0.85
*BRAF^V600E^*				
Mutated vs. Non-mutated	1.54	0.61	3.87	0.36
*APC*				
Mutated vs. Non-mutated	0.61	0.33	1.13	0.12
*TP53*				
Mutated vs. Non-mutated	3.20	1.31	7.80	0.01

**Table 4 cancers-14-03644-t004:** Multivariate survival model analysis for OS by mutation status of *APC*, *BRAF^V600E^*, *RAS*, *SMAD4*, and *TP53*.

*APC*, *BRAF^V600E^*, *RAS*, *SMAD4*, and *TP53* Mutation Status	Reference	Multivariable HR	95% CI	*p*
*SMAD4*-MT + *TP53*-MT	vs. rest of the population	2.5	1.44–4.36	0.001
*SMAD4*-MT + *RAS*-MT	vs. rest of the population	0.95	0.54–1.67	0.853
*SMAD4*-MT + *BRAF^V600E^*-MT	vs. rest of the population	1.5	0.64–3.47	0.353
*SMAD4*-MT + *APC*-MT	vs. rest of the population	0.51	0.29–0.91	0.022
*RAS*-MT + *TP53*-MT	vs. rest of the population	1.2	0.95–1.60	0.111

MT, mutated.

## Data Availability

The data that support the findings of this study are available from the corresponding author upon reasonable request.

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
