# Peer review of "Presence of Concurrent TP53 Mutations Is Necessary to Predict Poor Outcomes within the SMAD4 Mutated Subgroup of Metastatic Colorectal Cancer"

_cancers, 2022, doi:10.3390/cancers14153644_

Round 1

Reviewer 1 Report

This is very interesting study on a topic of great interest. However, I have some concerns:

1. Please clarify if some of the mCRC patients underwent curative intent surgery or not.

2. What was the statistical package you used for the analyses?

3. The multivariate analysis is the base that you support your conclusions. Thus, I suggest you explain the selection of factors you included in the univariate analysis. Why did you only include age, gender and sidedness? Since age and gender are weak- at best -prognosticators, your findings may not be fully adjusted for confounders. Indeed, according to suppl table 1 only tumor laterality was significant in univariate analysis and was included in multivariate.

4.  It is interesting that V600E BRAF mutation was not associated with worse prognosis even in univariate analysis. You had 32 cases so small number does not seem to be the reason for this unexpected finding. Can you please discuss that?

Author Response

This is very interesting study on a topic of great interest. However, I have some concerns:

  1. Please clarify if some of the mCRC patients underwent curative intent surgery or not.

Response: We do not have this data in our data base. However, one can make the argument that good prognostic biomarkers are more likely to have a response and to convert to surgery. Therefore, an imbalance in metastasectomies in favor non-SMAD4-TP53 mutant patients would not diminish from the significance of this biomarker.

  1. What was the statistical package you used for the analyses?

Response: For univariate and multivariate analysis, IBM SPSS Version 28.0.1.1 (15) was used. GraphPad Prism 9.3.1 was used for baseline characteristic table. We have added “IBM SPSS Version 28.0.1.1 (15) was used for univariate and multivariate analysis. GraphPad Prism 9.3.1 was used for baseline characteristic table” to line 99-101, page 5.

  1. The multivariate analysis is the base that you support your conclusions. Thus, I suggest you explain the selection of factors you included in the univariate analysis. Why did you only include age, gender and sidedness? Since age and gender are weak- at best -prognosticators, your findings may not be fully adjusted for confounders. Indeed, according to suppl table 1 only tumor laterality was significant in univariate analysis and was included in multivariate.

Response: To make sure our findings is adjusted for possible cofounders, we included as many variables as we possibly can in our univariate analysis. While age and gender did not make were not significant biomarkers in our univariate analysis, a wealth of data supports that elderly typically have a worse outcome than patients < 65. Similarly, several reports have shown interaction between gender and outcome with 5-FU based therapy. Therefore, we included these in our multivariate analysis based on historic data.

  1. It is interesting that V600E BRAF mutation was not associated with worse prognosis even in univariate analysis. You had 32 cases so small number does not seem to be the reason for this unexpected finding. Can you please discuss that?

Response: BRAFV600E was associated with worse OS in the univariate analysis of the whole population but lost its significance in the multivariate analysis, which suggest the prognostic value of BRAFV600E is affected by co-variables, including SMA4 and TP53. Indeed, these findings add more weight to our findings. It is also possibly that that BRAF-V600E may have been partially confounded by a small subpopulation with MSI (up to 30% of BRAF-V600E patients may be MSI). We have added “BRAFV600E was associated with worse OS in the univariate analysis of the whole population but lost its significance in the multivariate analysis, which suggest the prognostic value of BRAFV600E is affected by co-variables, including SMA4 and TP53. Indeed, these findings add more weight to our findings. It is also possibly that that BRAF-V600E may have been partially confounded by a small subpopulation with MSI (up to 30% of BRAF-V600E patients may be MSI).” to line 202-206, page 10.

Reviewer 2 Report

This study looked at the association of two genetic mutation, TP53 and SMAD4, coexisting together in patients with metastatic colorectal cancer can lead to poor overall survival. They have shown that SMAD4 alone cannot predict OS but when analysed together with TP53 mutation and negative OS can be predicted. Overall this is a very basic genomic study and the only merit of this paper is the clinical cohort that the readers had access too.

If this paper is to be published in this journal it would require the following amendments to be met before it is ready for publication. See below for suggested updates:

- Include a flow chart in the methods that show how the patient selection was made and how many samples were used for genomic analysis.

- In the methods for genomic analysis, it needs to be more detailed. Give more detailed description of how the analysis was done, what filtering, normalising procedures were used, were cofounding factors taken into consideration when doing the analysis. Since this is the main part of the paper, more details must be given.

- Since comprehensive genomic analysis was conducted in the tumour samples, it seems odd that the authors have not given some basic genomic analysis of the samples. A mutation landscape analysis of the tumour samples should be given to should the different mutation in the samples. Also along with the OS, signalling pathway analysis should also be shown.

Author Response

- Include a flow chart in the methods that show how the patient selection was made and how many samples were used for genomic analysis.

Response: Patients with mCRC treated at City of Hope Comprehensive Cancer Center between 2013 and 2020 with available tumor sequencing information via NGS by a CLIA assay were eligible for this study. We have included a flow chart in the method section. Please refer to page 5.  

- In the methods for genomic analysis, it needs to be more detailed. Give more detailed description of how the analysis was done, what filtering, normalising procedures were used, were cofounding factors taken into consideration when doing the analysis. Since this is the main part of the paper, more details must be given.

Response: The genomic profile of each tumor was extracted from NGS report of FDA approved diagnostic assay (FoundationOne, GEM ExTra, and Guardant360), so there was no filtering and normalization being done at our end. We have added “The genomic profile of each tumor was extracted from NGS report of these assays.” Please refer to line 97-98, page 6.

- Since comprehensive genomic analysis was conducted in the tumour samples, it seems odd that the authors have not given some basic genomic analysis of the samples. A mutation landscape analysis of the tumour samples should be given to should the different mutation in the samples. Also along with the OS, signalling pathway analysis should also be shown.

Response: we did not conduct the genomic sequencing by ourselves, the genomic profile of each tumor was extracted from NGS report of FDA approved diagnostic assay (FoundationOne, GEM ExTra, and Guardant360).

Reviewer 3 Report

The manuscript by Wang et al. assesses several potential prognostic biomarkers in a cohort of 433 patients with metastatic colorectal cancer through a multivariate survival analysis model. Coexisting mutations in TP53 and SMAD4 were found to be correlated with worse prognosis in metastatic colorectal cancer patients. This finding is welcome in light of the ongoing need for novel prognostic and predictive cancer biomarkers.

All work has been thoughtfully executed and properly discussed. However, there are a few issues that need to be addressed before publication:

-        In my opinion, the introduction should be more comprehensive; for instance, the authors could include a short paragraph on the incidence of metastatic colorectal cancer, survival rates etc.

-        Include in the Materials and Methods section a brief discussion regarding the treatment given to the patients, site of metastases, inclusion/exclusion criteria; even though the authors stated that this data is heterogenous throughout the patient population, I think that for full disclosure this information should be included in the paper

-        If possible, the authors should investigate the impact of other biomarkers (for instance microsatellite instability or oncogenes like TROP 2, PTEN, PIK3CA) on the overall survival of the population and/or in patients with SMAD4 mutations

-        Figure 1 and Supplementary Figures 1-3 contain a spelling mistake – colorectal is spelled ‘colorecal’

Author Response

  1. In my opinion, the introduction should be more comprehensive; for instance, the authors could include a short paragraph on the incidence of metastatic colorectal cancer, survival rates etc.

Response: I have added a short summary of the incidence, and overall survival of metastatic colorectal cancer. Please refer to page 4, line 62-64: “With 151,300 new cases and 52,580 deaths estimated in 2022, colorectal cancer remains to be the second most common cause of cancer related death in the United States. While localized disease has a 5-year overall survival rate of 90.9%, only 15% of patients with distant metastatic disease can survival for 5 years.”

  1. Include in the Materials and Methods section a brief discussion regarding the treatment given to the patients, site of metastases, inclusion/exclusion criteria; even though the authors stated that this data is heterogenous throughout the patient population, I think that for full disclosure this information should be included in the paper

Response: All patients were given standard of care, patients with available NGS report were included. We have added “All patients received standard of care” to line 90-91, page 5. In addition, we have added “Liver, lung, lymph node and peritoneal metastasis were found in 64.8% (n=281), 24.1% (n=104), 22.4% (n=97) and 17.8% (n=77) of the patients” to line 110-111, page 6.

  1. If possible, the authors should investigate the impact of other biomarkers (for instance microsatellite instability or oncogenes like TROP 2, PTEN, PIK3CA) on the overall survival of the population and/or in patients with SMAD4 mutations

Response: Given the limited resources, we were not able to investigate the impact of other biomarkers such as PTEN, PIK3CA, and MSI-H on the prognosis of the population.

  1. Figure 1 and Supplementary Figures 1-3 contain a spelling mistake – colorectal is spelled ‘colorecal’

Response: we have corrected those mistakes.

Round 2

Reviewer 1 Report

N/A

Reviewer 2 Report

No comments to add to the revised draft of the paper. All comments were addressed by the authors.

Reviewer 3 Report

The authors Wang et al. revised their manuscript ‘Presence of concurrent TP53 mutations is necessary to predict poor outcomes within the SMAD4 mutated subgroup of metastatic colorectal cancer’ to comply with most of my comments and suggestions. Therefore, I recommend publication of the manuscript.

Minor comments:

-        There are a couple of spelling mistakes: in the Scheme on page 2 - patients is spelled as ‘patietns’, and in line 218 – ‘SMA4’ should be SMAD4